# Micronutrient Intake during Complementary Feeding in Very Low Birth Weight Infants Comparing Early and Late Introduction of Solid Foods: A Secondary Outcome Analysis

**DOI:** 10.3390/nu16193279

**Published:** 2024-09-27

**Authors:** Melanie Gsoellpointner, Margarita Thanhaeuser, Margit Kornsteiner-Krenn, Fabian Eibensteiner, Robin Ristl, Bernd Jilma, Sophia Brandstetter, Angelika Berger, Nadja Haiden

**Affiliations:** 1Department of Neonatology, Kepler University Hospital, Johannes Kepler University, 4020 Linz, Austria; melanie.gsoellpointner@kepleruniklinikum.at; 2Department of Pediatrics and Adolescent Medicine, Comprehensive Center for Pediatrics, Medical University of Vienna, 1090 Vienna, Austria; margarita.thanhaeuser@meduniwien.ac.at (M.T.); margit.kornsteiner.krenn@gmail.com (M.K.-K.); fabian.eibensteiner@meduniwien.ac.at (F.E.); sophia.brandstetter@meduniwien.ac.at (S.B.); angelika.berger@meduniwien.ac.at (A.B.); 3Center for Medical Data Science, Medical University of Vienna, 1090 Vienna, Austria; robin.ristl@meduniwien.ac.at; 4Department of Clinical Pharmacology, Medical University of Vienna, 1090 Vienna, Austria; bernd.jilma@meduniwien.ac.at

**Keywords:** preterm infants, complementary feeding, micronutrient intake, iron, vitamin D, zinc, phosphorus, calcium

## Abstract

**Background/Objectives:** The complementary feeding period is crucial for addressing micronutrient imbalances, particularly in very low birth weight (VLBW) infants. However, the impact of the timing of solid food introduction on micronutrient intake in a representative VLBW population remains unclear. **Methods**: This prospective, observational study investigated micronutrient intake during complementary feeding in VLBW infants categorized based on whether solids were introduced early (<17 weeks corrected age (CA)) or late (≥17 weeks CA). Nutritional intake was assessed using a 24 h recall at 6 weeks CA and with 3-day dietary records at 12 weeks and at 6, 9, and 12 months CA. **Results**: Among 218 infants, 115 were assigned to the early group and 82 to the late group. In total, 114–170 dietary records were valid for the final analysis at each timepoint. The timepoint of solid introduction did not affect micronutrient intake, except for a higher iron and phosphorus intake at 6 months CA in the early group (early vs. late: iron 0.71 vs. 0.58 mg/kg/d, adjusted *p*-value (*p*-adj.) = 0.04; phosphorus 341 vs. 286 mg/d, *p*-adj. = 0.04). Total vitamin D, calcium, zinc, and phosphorus greatly met intake recommendations; however, dietary iron intake was insufficient to equalize the iron quantity from supplements during the second half year CA. While nutrient intakes were similar between infants with and without comorbidities, breastfed infants had lower micronutrient intakes compared with formula-fed infants. **Conclusions**: This study suggests that micronutrient intakes were sufficient during complementary feeding in VLBW infants. However, prolonged iron supplementation may be necessary beyond the introduction of iron-rich solids. Further research is essential to determine micronutrient requirements for infants with comorbidities.

## 1. Introduction

The complementary feeding phase is acknowledged as a critical period for addressing micronutrient imbalances, which may be even more important to very low birth weight (VLBW) infants [1]. For term infants, the European Society for Pediatric Gastroenterology, Hepatology, and Nutrition (ESPGHAN) recommends introducing solids between the 17th and 26th week of life [2]. However, evidence-based guidelines for preterm infants are still lacking. Observational studies indicate that preterm infants are introduced to solid foods at an earlier developmental stage than full-term infants [3,4].

Micronutrients such as iron, vitamin D, calcium, phosphorus, and zinc are of particular interest in VLBW infants due to diminished body stores, elevated micronutrient needs for catch-up growth, for example, and the nutrients’ critical roles in developmental processes [5,6].

Although some studies have investigated micronutrient intake during early postnatal phases, only a few studies are available regarding micronutrient intake during the complementary feeding period in VLBW infants, with most of them being outdated [7,8]. Notably, a recently published two-arm randomized interventional study investigated an early (between 10 and 12 weeks corrected age (CA)) versus late (between 16 and 18 weeks CA) introduction of solid foods in VLBW infants. Infants were fed a standardized, age-dependent, step-up feeding regimen until 12 months CA [9,10]. While this study made significant strides in elucidating micronutrient intake during complementary feeding in preterm infants, its exclusion of infants with comorbidities that may impede stable growth restricts its applicability to a broader VLBW population. Furthermore, the adequacy of current non-standardized micronutrient intakes and the potential influence of the timing of solids introduction, considering term infant guidelines (before vs. at or above 17 weeks CA) on critical micronutrient intakes, have yet to be determined.

Thus, this secondary analysis of a prospective observational study aims to investigate micronutrient intake patterns during complementary feeding in former VLBW infants, comparing the introduction of solids before and after 17 weeks CA. This study includes infants with preterm-birth-related comorbidities such as bronchopulmonary dysplasia (BPD), necrotizing enterocolitis (NEC), and intraventricular hemorrhage (IVH). Our objective was to gain a better understanding of complementary feeding practices in a representative VLBW cohort and to assess micronutrient intake in those with comorbidities. Additionally, this study aims to provide nutritional intake data comparing breastfed, formula-fed, and mixed-fed infants during the first year of life.

## 2. Materials and Methods

### 2.1. Study Design

This study constitutes a secondary analysis of nutritional intake data obtained from a prospective, observational study in VLBW infants. Infants with a birth weight below 1500 g and a gestational age < 32 weeks were included in the study. Exclusion criteria encompassed congenital heart disease, chromosomal aberrations, major congenital birth defects, and Hirschsprung disease. Study subjects were longitudinally monitored in the outpatient clinic of the Division of Neonatology at the Medical University of Vienna between April 2016 and November 2021.

Based on the parental decision to start complementary feeding, infants were categorized into either an early (<17th week CA) or late (≥17th week CA) complementary feeding group.

The study received ethical approval from the ethics committee of the Medical University of Vienna (EK: 1273/2016) and was registered on clinicaltrials.gov (NCT02936219). Informed consent was obtained from at least one parent or legal guardian for each participating infant. The primary outcome report of this study was previously published by Thanhaeuser et al. [11].

### 2.2. Study Visits, Nutrient Intake Data Collection, and Evaluation

Infants were regularly followed up in the neonatal outpatient clinic of the Division of Neonatology. Within regular checkups, study visits were conducted covering term, 6 weeks, 12 weeks, and 6, 9, and 12 months CA. At the respective timepoints, standardized procedures for the collection of anthropometric measurements, including body weight, length, and head circumference, were conducted. Parents of participating infants underwent guidance from a nutritionist, equipping them to maintain food logs that documented enteral intake at specified intervals. At 6 weeks CA, parents were instructed to complete a 24 h recall, while 3-day dietary records were required at 12 weeks and at 6, 9, and 12 months CA. While 24 h recalls yielded retrospective information on foods and beverages consumed on a given day, prospective 3-day dietary records consisted of three consecutive days, including one weekend day. For both dietary assessments, parents were asked to include details on product brands and recipes of self-prepared meals.

The subsequent analysis of dietary records was undertaken by a nutritionist using nut.s nutritional software, referencing the German nutrient database and the Austrian nutrient table (nut.s nutritional software, Vienna, Austria, Version II.3.1).

For infants who were breastfed, estimated average values for consumed mother’s milk, as published by Dewey et al., were utilized [12]. Detailed information on infant formula had to be documented in each protocol. To ensure precise nutrient analysis, recipes of all infant formulas were sought from distinct manufacturers, and changes in formulations were factored into the calculations. Body weight at the corresponding date of the dietary record was employed to calculate protein intake per kilogram of body weight. For months without measurements, body weight was determined based on the daily increase between the closest two measurements.

Prior to March 2017, infants received a supplementation of 650 international units (IU)/d of vitamin D3 until reaching one year CA. However, in April 2017, this protocol was revised to 800 IU/d of vitamin D3 supplementation until one year CA. Moreover, infants were supplemented with 2–3 mg/kg/d iron (administered as Ferrum Hausmann^®^, iron oxide polymaltose complex, Vifor France, Paris, France) until a regular meat intake was established. Precise dosages, as well as the initiation and cessation dates of all supplements, were recorded and utilized for daily intake computation. Dietary intake refers exclusively to oral consumption from foods, while total intakes encompass both dietary and supplementary intake.

### 2.3. Outcome Data

The primary outcome of this analysis was iron intake (mg/kg/d) at 6 and 12 weeks and at 6, 9, and 12 months CA. Secondary outcomes included intake of total iron, dietary vitamin D, total vitamin D, calcium, phosphorus, and zinc. Additionally, we performed a subgroup analysis to compare micronutrient intakes between infants with BPD, NEC, or IVH and infants without comorbidities. Moreover, we compared micronutrient intakes in breastfed, formula-fed, and mixed-fed infants.

Nutrient intakes were compared with current dietary intake guidelines for preterm infants (iron, vitamin D). For zinc, phosphorus, and calcium, no specific dietary intake recommendations for preterm infants were available; hence, dietary intake recommendations for term infants were used for comparison.

### 2.4. Sample Size Calculation

The sample size calculation for this study was based on the assumption that a 5% difference in body length represents the minimally clinically relevant effect. A coefficient of variation of 11% was assumed to estimate the standard deviation, with the significance level set at 0.05. Using these parameters, along with a power of 80%, a sample size of 152 infants (76 per group) was calculated to detect a statistically significant difference in length gain between the early and late observation groups. To account for an anticipated 30% dropout rate, the final sample size required was 198 infants.

### 2.5. Statistical Analysis

Assessment of nutrient intake involved a comparative analysis between the early and late introduction of solid foods (before and after 17 weeks CA). Participants who were lost to follow-up, withdrew informed consent, or had insufficient dietary records were excluded from this secondary outcome analysis. However, dietary protocols up to the point of lost follow-up or withdrawal of informed consent, as well as protocols of infants without the primary outcome of the study, were included in this analysis.

Neonatal baseline characteristics are provided as mean with standard deviation (SD). To detect differences between baseline characteristics, the chi-squared test or the Mann–Whitney U-test was applied.

Statistical analysis of the primary and secondary outcomes employed linear mixed-effects models. These models incorporated relevant covariates such as group assignment (early vs. late introduction), sex, gestational age, and nutrition at 6 weeks CA (breastmilk, formula, or mixed feeding). A random intercept was used to account for potential correlations among siblings of multiple births. We conducted subgroup analyses to investigate micronutrient intake by comparing infants with and without comorbidities (BPD, NEC ≥ grade II, IVH ≥ grade II). BPD was defined as oxygen demand ≥ 36 + 0. Diagnosis of NEC was made by clinical and radiological findings according to Bell’s staging criteria for NEC. [13] The degree of IVH was graded I through IV according to Papile et al. [14].

Additional subgroup analyses focused on nutritional intake based on the type of milk feeding (breastfed vs. formula-fed, breastfed vs. mixed-fed). Mixed feeding is defined as the consumption of both breastmilk and infant formula. For both subgroup analyses, either a Student’s t-test or the Mann–Whitney U test was used. Correlations between siblings of multiple births were not considered for subgroup analyses due to the smaller subgroup sample sizes.

Marginal means and means (subgroups), accompanied by corresponding standard errors (SE), were computed, with *p*-values used to evaluate the null hypothesis of no between-(sub)group differences. Graphical depictions illustrated estimated marginal means and standard errors through error bars, with significance set at *p*-values < 0.05. An additional analytical approach involved adjusting *p*-values for comparisons between (sub)groups for the same nutrient at distinct timepoints using the Bonferroni-Holm method (*p*-adj). All statistical analyses were performed using R version 4.1.1 (R Core Team, 2022).

## 3. Results

### 3.1. Study Participants, Baseline Characteristics, and Dietary Records

A total of 529 infants met the inclusion criteria for this study. However, in 311 cases, either the parents declined participation or had already introduced solid foods. As a result, 218 infants were ultimately enrolled in the study. After excluding 21 infants due to withdrawal of parental consent (n = 2), unavailability of data on the beginning of complementary feeding (n = 4), loss of follow-up (n = 4), screening failure (n = 1), or missing data on the primary outcome of the study (n = 10), the cohort for the initial primary outcome analysis included 197 infants [11].

Out of the initial cohort, 115 patients were allocated to the early complementary feeding group and 82 infants to the late group. Dietary records that were incomplete or inaccurately documented were excluded from the analysis. However, records from participants who were lost to follow-up, withdrew consent, or did not have missing data on the primary outcome were retained. This resulted in 114–170 valid dietary records for analysis at each timepoint, as shown in Table 1.

Infants in the early introduction group began solid food consumption at a mean age of 13.2 weeks CA (SD ± 3.0), while infants in the late group were introduced to solids at a mean age of 20.4 weeks CA (SD ± 2.9). The early introduction group had a slightly higher mean gestational age (188 days, SD ± 14) and birthweight (926 g, SD ± 254) compared with the late introduction group (187 days, SD ± 17; 881 g, SD ± 262). However, these baseline differences were not statistically significant. While infants with IVH and NEC were evenly distributed between the early and late groups, the prevalence of infants with BPD was higher in the late group (12% vs. 28%, *p* = 0.01) [11] (Table 2).

### 3.2. Micronutrient Intake Comparing Early Versus Late Introduction of Solid Foods

#### 3.2.1. Dietary and Total Iron Intake

Dietary iron intakes were slightly higher in the early group until 9 months CA, with a significant difference observed at 6 months CA (early: 0.71 mg/kg/d (SE ± 0.03), late: 0.58 mg/kg/d (SE ± 0.04); *p*-adj: 0.04). Generally, dietary iron intakes were low, with mean intakes ranging from 0.64 to 0.74 mg/kg/d in the early group and 0.55 to 0.75 mg/kg/d in the late group (Figure 1A).

Moreover, absolute dietary iron intakes (mg/day) did not exceed 6.3 mg/day throughout the observation period, falling short of the recommended intake of 11 mg/day for term infants aged 4 to 12 months [15] (see Appendix A).

Overall, mean total iron intakes (dietary + supplementation) were similar between both groups. Total iron intakes peaked at 6 weeks CA [early: 4.15 mg/kg/d (SE ± 0.16), late: 4.33 mg/kg/d (SE ± 0.16); *p* = 0.56], but steadily decreased to values below the ESPGHAN recommendations (2–3 mg/kg/d until 6–12 months of age) at 9 and 12 months CA in both study groups (9 months CA—early: 1.27 mg/kg/d (SE ± 0.16), late: 1.55 mg/kg/d (SE ± 0.17), *p* = 0.56; 12 months CA—early: 1.00 mg/kg/d (SE ± 0.15), late: 1.48 mg/kg/d (SE ± 0.16), *p* = 0.15) [5] (Figure 1B).

#### 3.2.2. Dietary and Total Vitamin D Intake

The mean dietary vitamin D intake was similar between both groups throughout the observation period, consistently remaining low without any discernible trend of change. In the early group, mean dietary vitamin D intake ranged from 264 to 328 IU/d, while in the late group it ranged from 255 to 285 IU/d (Figure 2A).

At 6 weeks CA, mean total vitamin D intakes (dietary + supplementation) were higher in the early group, although this difference was not statistically significant (early group: 1116 IU/d (SE ± 39), late group: 1011 IU/d (SE ± 39); *p*-adj. = 0.17). Overall, total vitamin D intakes were similar between both groups, with levels in the early group ranging from 986 to 1016 IU/d, and in the late group from 993 to 1012 IU/d, from 12 weeks until 9 months CA. The late group consistently met the ESPGHAN intake recommendations (800–1000 IU/d) at all investigated timepoints. In contrast, mean total vitamin D intake in the early group was below the recommendations at 12 months CA (692 IU/d (SE ± 55)) [16] (Figure 2B).

#### 3.2.3. Dietary Calcium and Phosphorus Intake

Mean dietary calcium intake was similar among the groups throughout the complementary feeding period. Mean calcium intakes increased over the observation period, rising from 373 to 551 mg/d in the early group and from 254 to 537 mg/d in the late group. Currently, there are no explicit reference values established for preterm infants. When comparing dietary intake levels against recommendations for term infants (0–3 months: 220 mg/d, 4–12 months: 330 mg/d), the findings demonstrate that these intakes were consistently met throughout the first year of life [15] (Figure 3A).

The early group demonstrated marginally higher mean dietary phosphorus intakes throughout the observation period. However, statistical significance was achieved only at 6 months CA (early: 341 (SE ± 14) mg/d, late: 286 (SE ± 15); *p*-adj. = 0.04). As of now, there are no specific dietary intake recommendations for phosphorus in VLBW infants. Hence, we compared our results with current term infant recommendations (0–3 months: 120 mg/day, 4–12 months: 180 mg/day), revealing that phosphorus intakes were adequate at all investigated timepoints [15] (Figure 3B).

#### 3.2.4. Dietary Zinc Intake

The mean dietary zinc intake was higher in the early group until 6 months CA; however, no statistical significance was observed after adjusting for multiple testing (6 weeks CA—early: 3.3 (SE ± 0.2) mg/d, late: 2.8 (SE ± 0.2), *p*-adj. = 0.07; 12 weeks CA—early: 3.4 (SE ± 0.2) mg/d, late: 2.8 (SE ± 0.2), *p*-adj. = 0.06; 6 months CA—early: 4.1 (SE ± 0.2) mg/d, late: 3.5 (SE ± 0.2), *p*-adj. = 0.06).

Dietary zinc recommendations for term infants (0–3 months: 1.5 mg/d; 4–12 months: 2.5 mg/d) were met throughout the observation period [15].

### 3.3. Micronutrient Intake in Infants without Comorbidities and Infants with BPD, NEC, or IVH

While dietary iron intakes were similar among infants with and without comorbidities (without comorbidities: 0.63–0.73 mg/kg/d; BPD: 0.62–0.92 mg/kg/d; NEC: 0.55–0.77 mg/kg/d; IVH: 0.57–0.81 mg/kg/d), there was a tendency toward higher total iron intakes in infants with comorbidities compared with those without comorbidities. This was most pronounced in infants with NEC at the end of the first year CA (6, 9, 12 months CA with no comorbidity: 2.34, 1.00, 1.01 mg/kg/d; NEC: 3.51, 2.54, 1.80 mg/kg/d; *p*-adj = 0.05, 0.05, 0.10). The ESPGHAN recommendations for iron intake (2–3 mg/kg/d until 6–12 months of age) were met until 6 months CA in infants with IVH, BPD, and without comorbidities, and until 9 months CA in infants with NEC (Figure 4A).

Dietary vitamin D intake was similar among the subgroups, with ranges as follows: no comorbidity: 266–354 IU/d; BPD: 263–302 IU/d; NEC: 284–388 IU/d; IVH: 224–359 IU/d. There was also no significant difference in total vitamin D intake between the subgroups. Total vitamin D intakes were adequate until 9 months CA in all groups. However, infants with NEC and IVH had total vitamin D intakes below the recommended levels (800–1000 IU/d) at 12 months CA (NEC: 740 IU/d (SE ± 146), IVH: 719 IU/d (SE ± 111)) (Figure 4B).

Similar dietary intakes were observed among all subgroups for calcium and zinc (mg/d), with both groups meeting current recommendations at all investigated timepoints. (Figure 4C,E) There was no difference in phosphorus intake between the subgroups until 9 months CA. However, at 12 months CA, phosphorus intake was significantly lower in infants with NEC compared with those without comorbidities (no comorbidity: 548 mg/d (SE ± 17), NEC: 397 mg/d (SE ± 35); *p* = 0.04). Despite this difference, phosphorus intake recommendations were met at all investigated timepoints in all groups (Figure 4D).

### 3.4. Micronutrient Intake in Breastfed, Formula-Fed, and Mixed-Fed Infants

Breastfed infants had a significantly lower mean dietary iron intake compared with formula-fed and mixed-fed infants throughout the observation period. Mean intakes in breastfed infants ranged from 0.09 to 0.52 mg/kg/d, whereas levels in the formula-fed group ranged from 0.77 to 0.99 mg/kg/d and in mixed-fed infants from 0.55 to 0.89 mg/kg/d (Figure 5A). Moreover, total iron intake was significantly lower in breastfed compared with formula- and mixed-fed infants at 6 weeks CA (breastfed: 3.72 mg/kg/d (SE ± 0.15), formula: 4.49 mg/kg/d (SE ± 0.18), *p*-adj = 0.003; mixed: 4.62 mg/kg/d (SE ± 0.28), *p*-adj = 0.007) and compared with formula-fed infants at 9 months CA (breastfed: 0.65 mg/kg/d (SE ± 0.13), formula: 1.39 mg/kg/d (SE ± 0.13); *p*-adj < 0.001). The ESPGHAN recommendations for iron intake (2–3 mg/kg/d) were not met at 9 and 12 months CA, regardless of the type of milk feeding (Figure 5B).

Dietary vitamin D intakes were significantly lower in breastfed infants compared with both formula- and mixed-fed infants throughout the observation period (breastfed: 54–199 IU/d, formula: 313–435 IU/d, *p*-adj < 0.001 for all; mixed: 244–371 IU/d, *p*-adj = 0.002 at 6 weeks CA, < 0.001 at 12 weeks and 6 months CA). Similarly, total vitamin D intake was significantly lower in breastfed infants compared with formula-fed infants until 9 months CA and compared with mixed-fed infants until 6 months CA. (Figure 5C) While formula-fed infants consistently met ESPGHAN intake recommendations, breastfed infants failed to meet them at 9 and 12 months CA (breastfed: 799 IU/d (SE ± 34) at 9 months CA, 714 IU/d (SE ± 98) at 12 months CA) (Figure 5D).

Furthermore, significantly lower intakes of calcium, phosphorus, and zinc were observed in breastfed infants compared with formula- and mixed-fed infants at all observed timepoints, except for phosphorus at 12 months CA. While formula- and mixed-fed infants surpassed the daily intake recommendations for these nutrients, breastfed infants did not meet the recommendations at 6 months for calcium (recommendations: 0–3 months: 220 mg/d; 4–12 months: 330 mg/d; breastfed: 302 mg/d) and at 12 weeks and 6 months for zinc (recommendations: 0–3 months: 1.5 mg/d; 4–12 months: 2.5 mg/d; breastfed: 1.21 mg/d and 1.91 mg/d, respectively) [15] (Figure 5E–G).

## 4. Discussion

This secondary outcome analysis of a prospective observational study investigated critical micronutrient intake during the first year of life, comparing an early and late introduction of solids in VLBW infants. To our knowledge, this was the first trial reporting data on micronutrient intake patterns during complementary feeding in a representative VLBW cohort including those with preterm-birth-related comorbidities.

Overall, the timepoint of the introduction of solids did not have an impact on mean dietary iron intake, except for a higher iron and phosphorus intake at 6 months CA in the early group. Mean total iron intakes were adequate until 6 months CA; however, recommendations were not met during the second half year CA in both groups. Total vitamin D intakes met current intake recommendations in VLBW infants during the first year of life. Mean intakes of calcium, zinc, and phosphorus were slightly higher in the early group but were shown to be sufficient in both complementary feeding groups when compared with term infant recommendations. This study further revealed that micronutrient intakes were similar between infants with and without comorbidities. However, breastfed infants had significantly lower intakes of most micronutrients compared with formula- and mixed-fed infants.

### 4.1. Dietary and Total Iron Intake

This study demonstrated that infants introduced to solids early had similar mean dietary iron intakes compared with those introduced later, except for a higher intake observed at 6 months CA. Marriot et al. reported significantly higher iron intakes (9.2 vs. 6.7 mg/d) at 12 months CA in preterm infants that were introduced to solids early (early: 13 weeks uncorrected age, late: 17 weeks uncorrected age). Parents of infants that were in the early introduction group were specifically advised to provide foods that are rich sources of iron [8]. However, this study seems to be no longer applicable, as it was conducted at a time when post-discharge formula was not available. Similarly, Kattelman et al. found that an early introduction of solids led to higher dietary iron intakes (16.2 vs. 13.2 mg/day), partly due to the introduction of iron-fortified cereals as the initial solid food [17]. In contrast, a randomized intervention trial comparing early (10–12 weeks CA) versus late (16–18 weeks CA) introduction of solids in VLBW infants using a standardized step-up complementary feeding approach found similar iron intakes between both groups throughout the first year of life [10].

In our study, breastfed infants had a significantly lower mean dietary iron intake compared with formula-fed infants (0.26 vs. 0.77 mg/kg/d; *p* < 0.001). Thus, we believe that the higher intake observed in the early introduction group at 6 months CA in our study most likely resulted from higher intakes in formula-fed infants and lower breastfeeding rates in the early introduction group.

Iron intake levels remained consistently low, with a maximum mean intake of 0.75 mg/kg/day. Notably, dietary iron intake showed only minimal increases throughout the observation period, even when meat meals and iron-rich solids were regularly included. Additionally, absolute dietary iron intakes (mg/day) did not exceed 6.3 mg/day, falling short of the recommended intake of 11 mg/day for term infants aged 4 to 12 months CA during the observation period [15]. Accordingly, introducing meat or iron-rich foods did not significantly enhance iron intake and may not sufficiently replace supplementation.

Iron deficiency can have long-lasting effects on neurodevelopment. Studies show that iron deficiency during infancy is associated with cognitive impairment, attention deficits, lower educational performance, and poorer emotional health during adulthood, highlighting the critical need for adequate iron intake to support healthy development [18,19].

The ESPGHAN recommends iron supplementation of 2–3 mg/kg/day until 6–12 months, depending on the infant’s diet [5]. Our data indicated that iron supplementation was appropriate until 6 months CA. However, it significantly decreased afterwards, falling short of the recommendations during the second half of the first year CA. Given that the dietary iron intake (mg/kg/d) did not sufficiently increase at the time when supplementation decreased, we suggest that iron supplementation intake may be prolonged. Special attention should be drawn to breastfed infants, as total iron intakes were significantly lower at 9 months CA compared with formula-fed infants, with levels decreasing to 0.65 mg/kg/d. The data presented in this study suggest that current clinical guidelines for iron supplementation may require revision. First, the discontinuation of iron supplementation should be guided by monitoring iron status rather than solely by the introduction of regular meat and iron-rich meals. Additionally, iron supplementation guidelines may need to be tailored specifically for breastfed and formula-fed infants to ensure appropriate management of iron status in each group.

This study further included infants with preterm-birth-related comorbidities such as BPD, NEC, or IVH. Chronic stress, inflammation, heightened respiratory effort, and lung damage may increase nutrient demands in infants with BPD [20,21]. We found similar dietary and total iron intakes in infants with and without BPD. However, it remains unclear whether this is sufficient to cover the needs of infants with BPD. Karatza et al. suggests that infants with BPD require iron intakes of 4 mg/kg/day until 12 months of age [22]. Our study indicated that these values were not met beyond 6 weeks CA in infants with BPD, suggesting that those infants were at a high risk of iron deficiency. Unfortunately, this study did not investigate iron status; thus, we are unaware whether iron deficiency was more prevalent in infants with BPD. Furthermore, infants with NEC may be at risk for malabsorption and limited nutrient intake, necessitating higher nutritional intakes in the long term [23]. We observed similar dietary iron intakes but higher total iron intakes in infants with NEC compared with those without, likely due to increased supplementation. Data on iron requirements for VLBW infants with NEC during complementary feeding are lacking, making it difficult to assess iron intake adequacy. Thus, studies are needed to investigate iron requirements in infants with comorbidities during the first year of life.

In general, iron intake of VLBW infants can be further optimized, particularly during the later stages of complementary feeding. This can be achieved through prolonged iron supplementation, nutritional guidance, and educating parents about iron-rich dietary sources. Previous studies have shown that iron-fortified cereals effectively increase iron intake and reduce the risk of iron deficiency [24,25]. Introducing iron-fortified cereals as one of the first solids may improve iron status, especially in at-risk infants.

### 4.2. Dietary and Total Vitamin D Intake

We found no significant difference in mean dietary vitamin D intake between the early and late feeding groups. Mean vitamin D intakes were consistently low, ranging between 255 and 328 IU/day. It is widely acknowledged that dietary sources alone are insufficient to meet vitamin D requirements [26]. While vitamin D can be synthesized through sun exposure, infants need protection from sunlight, making them reliant on enteral vitamin D supply [27]. Therefore, the ESPGHAN recommends vitamin D supplementation of 800–1000 IU/day until 12 months of age in VLBW infants [16].

Total vitamin D intake was similar among both groups, and ESPGHAN recommendations were met until 9 months CA in the early group and throughout the observation period in the late group. However, our study indicated that vitamin D supplementation decreased before reaching 12 months CA in both groups. Advocating for continued vitamin D supplementation until 12 months CA could enhance vitamin D status and reduce the risk of deficiency. This is especially important because previous studies have shown a high prevalence of deficiency in VLBW infants at 12 months CA (81–89%). [28] Therefore, as emphasized by Thanhaeuser et al., current recommendations for vitamin D supplementation in preterm infants may need to be revised, and regular monitoring of vitamin D status to tailor supplementation individually would be beneficial in preventing and treating low vitamin D levels [28]. 

Dietary and total vitamin D intakes were similar in infants with BPD, NEC, and IVH compared with those without these comorbidities. However, infants with NEC (740 IU/d) or IVH (719 IU/d) fell short of recommendations at 12 months CA.

Low vitamin D levels have been shown to increase the risk of NEC in preterm infants [29]. Therefore, infants diagnosed with NEC may already have lower baseline vitamin D levels. Furthermore, malabsorption of vitamin D may occur in infants with NEC, especially in those who have undergone bowel resection, thereby heightening nutritional requirements. The severity of NEC and associated gastrointestinal complications can further impair the absorption and utilization of vitamin D, putting these infants at high risk of vitamin D deficiency [30]. Thus, ensuring adequate vitamin D supplementation until at least 12 months CA or even longer is particularly crucial. However, further studies are warranted to define vitamin D requirements in infants with varying severity grades of NEC over the long term.

Additionally, a special focus should be given to breastfed infants, as mean total vitamin D intakes were at the borderline of meeting recommendations from 12 weeks CA onwards (714–811 IU/d). Breastmilk contains only a little vitamin D, and the nutritional supply from solids is minimal. Við Streym and colleagues have shown that breastfed term infants receive less than 20% of the daily recommended vitamin D intake from breastmilk [31]. As the vitamin D requirements of preterm infants are higher, the proportional amount that can be covered from supplied breastmilk is even lower [28]. Thus, vitamin D supplementation in these infants should be promoted, and parents should be educated on the importance of compliance with vitamin D supplementation until 12 months CA. Regular monitoring in high-risk patients offers individually tailored supplementation and the prevention of abnormal bone health. Current vitamin D guidelines do not differentiate between breastfed and formula-fed infants. However, the data presented in this study suggest that such distinctions may be necessary to optimize vitamin D intake, particularly for breastfed infants.

### 4.3. Dietary Calcium and Phosphorus Intake

Appropriate levels of calcium and phosphorus are crucial for optimal bone health. Insufficient intake of these minerals can lead to inadequate osteoid mineralization and reduced bone mineral content, potentially resulting in rickets, especially in premature infants [32]. Thus, both calcium and phosphorus are crucial in preventing metabolic bone disease, highlighting the need for adequate dietary intakes [33].

Our study showed that the timepoint of solid food introduction did not affect calcium intake. Additionally, both groups met the recommended dietary calcium intake for term infants (calcium: 0–3 months: 220 mg/day; 4–12 months: 330 mg/day) at all investigated timepoints [15]. This is in line with other studies that reported similar calcium intakes from 3 to 12 months CA (342–520 mg/d), exceeding current intake recommendations [10,28]. Moreover, phosphorus intake was similar between the early and late introduction groups, except for 6 months CA, where the early group had significantly higher intakes (*p* = 0.01). Formula-fed infants had significantly higher phosphorus intakes until 9 months CA compared with breastfed infants. Consequently, the higher phosphorus intake in the early group potentially resulted from the higher intake and the increased prevalence of formula-fed infants in the early group. Despite these differences, phosphorus intakes met the guidelines for term infants (0–3 months: 120 mg/day; 4–12 months: 180 mg/day) throughout the study in both groups [15]. However, it remains to be determined if the recommendations for term infants can be directly applied to preterm infants, particularly those with long-term health conditions.

Infants with BPD and IVH had similar calcium and phosphorus intakes compared with those without these conditions, but those with NEC had a significantly lower phosphorus intake at 12 months CA compared with infants without comorbidities. This study did not evaluate bone health; hence, we are unable to conclude whether the observed intakes are adequate for infants with critical illness. Randomized controlled trials are essential to determine adequate dietary calcium and phosphorus intakes during this critical period for infants with comorbidities.

We found that breastfed infants had significantly lower calcium and phosphorus intakes throughout the observation period compared with formula-fed infants. Additionally, breastfed infants did not meet calcium intake recommendations at 6 months CA (227 ± 13 mg/d). Inadequate calcium intake can result in stunted growth and decreased bone mineral content [34]. To decrease the risk of calcium deficiency, solids high in bioavailable calcium, such as dairy products, beans, broccoli, and spinach, should be offered [35].

### 4.4. Dietary Zinc Intake

Zinc plays a crucial role as a trace element, impacting growth, tissue differentiation, and immune function [36]. However, preterm infants face an increased risk of zinc deficiency due to their low body stores at birth [37].

Our study found that an early introduction of solid foods resulted in marginally higher dietary zinc intake at both 6 weeks and 6 months CA compared with late introduction. While this hints at a potential increase in zinc intake through early solid food introduction, a recent randomized trial in VLBW infants did not definitively show the benefits of early introduction in improving zinc intake [10]. The higher zinc intake in the early introduction group in our study may stem from an earlier transition to zinc-rich foods like meats and fish [38] or from lower zinc intakes in breastfed infants and the higher breastfeeding rate in the late group (breastfed: 1.21–4.05 mg/d; formula-fed: 4.3–5.24 mg/d).

The recorded zinc intakes align with previous studies, but specific recommendations for VLBW infants during complementary feeding are lacking [10,17]. Comparing our results with term infant recommendations, set at 1.5 mg/day for ages 0–3 months and 2.5 mg/day for ages 4–12 months, both groups met the recommended intake levels for term infants at all investigated timepoints [15]. However, it remains uncertain if these intakes suffice during the complementary feeding period, especially in preterm infants with comorbidities that may affect nutrient intake and absorption or who require higher needs due to long-term disease burden. This gap persists, as we did not assess zinc status, and other trials on VLBW infants are lacking. Despite this, advocating for zinc-rich solid foods is presumed to be beneficial for preterm infants [1]. In particular, this applies to exclusively breastfed infants, considering that zinc intake recommendations were not met at 6 months CA in breastfed infants as well as the declining zinc content in human milk postpartum [39].

Future studies are crucial to ascertain if zinc intake recommendations for term infants apply to VLBW infants or if this population requires specific guidelines or supplementation.

### 4.5. Strength and Limitation

This secondary analysis of a prospective observational trial yields significant insights into complementary feeding practices among preterm infants, offering valuable perspectives on their micronutrient intakes during the critical first year of life. The inclusion of a diverse cohort of VLBW infants with comorbidities enhances the generalizability of the findings. Nevertheless, the study’s reliance on average breastmilk intake data constitutes a notable limitation. Additionally, the trial was not powered to detect significant differences in nutritional intake between the study groups. The number of infants in the subgroups (type of feeding, comorbidities) was limited; therefore, these results should be interpreted cautiously. Furthermore, the observational design of the study, while reducing its quality in terms of causal inference, facilitates the provision of valuable information on current nutrient intakes in a representative population of Austrian VLBW infants.

## 5. Conclusions

In conclusion, this observational study indicates that the timing of solid introduction was not associated with micronutrient intakes during the first year of life. Dietary intakes of zinc, calcium, and phosphorus were adequate. However, dietary intakes of iron were insufficient to equal the iron quantity from supplements; hence, prolonged iron supplementation may be necessary beyond the introduction of meat and iron-rich solids. A decrease in total vitamin D intake by the end of the first year CA calls for improved adherence to vitamin D supplementation until 12 months CA. Although micronutrient levels were similar between infants with and without NEC, BPD, or IVH, it remains to be determined whether current dietary intakes are adequate for these infants. This underscores the need for future research to establish specific micronutrient requirements for these subgroups. Additionally, special attention should be given to breastfed infants, as they are at the highest risk for micronutrient deficiencies

## Figures and Tables

**Figure 1 nutrients-16-03279-f001:**
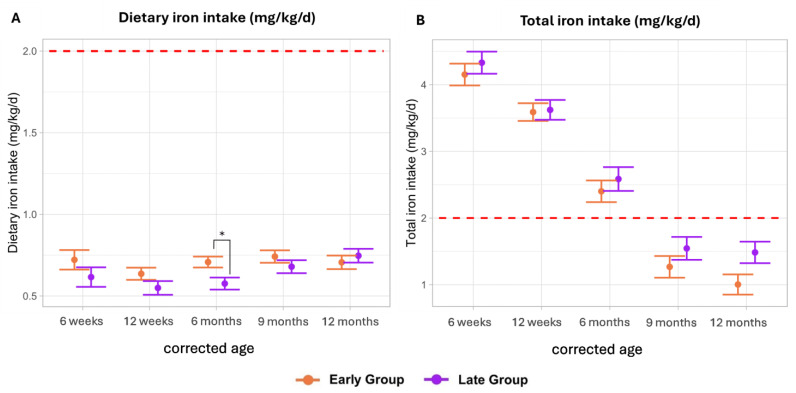
Dietary and total iron intake (mg/kg/d). Data are presented as the estimated marginal mean and standard error. Significant differences (adjusted *p*-value < 0.05) are marked with *. The red dotted lines represent the recommended iron supplementation (2–3 mg/kg/d until 6–12 months) in preterm infants. Total = dietary intake + supplementation. (**A**) Dietary iron intake (mg/kg/d) comparing an early vs late introduction of solid foods. (**B**) Total iron intake (mg/kg/d) comparing an early vs late introduction of solid foods.

**Figure 2 nutrients-16-03279-f002:**
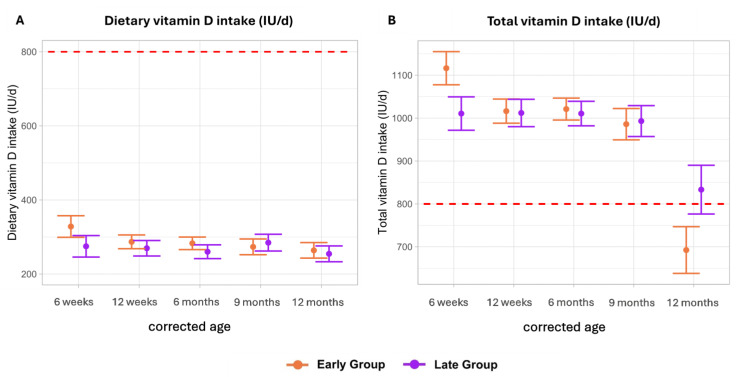
Dietary and total vitamin D intake (IU/d). Data are presented as the estimated marginal mean and standard error. The red dotted lines represent the recommended vitamin intake (800 IU/d) in preterm infants. Total = dietary intake + supplementation. (**A**) Dietary vitamin D intake (IU/d) comparing an early vs late introduction of solid foods. (**B**) Total vitamin D intake (IU/d) comparing an early vs late introduction of solid foods.

**Figure 3 nutrients-16-03279-f003:**
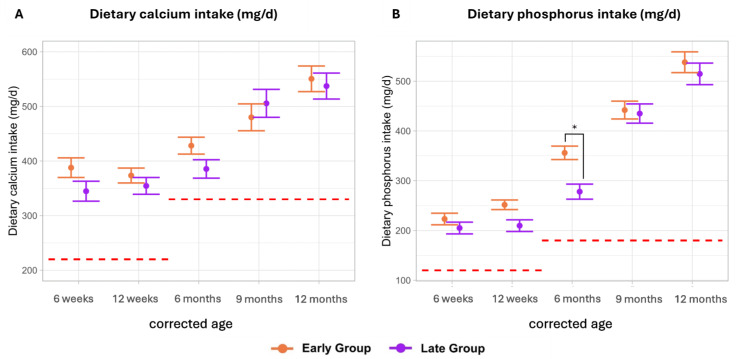
Dietary calcium and phosphorus intake (mg/d). Data are presented as the estimated marginal mean and standard error. The red dotted lines represent the recommended intake of (**A**) calcium (0–3 months: 220 mg/d, 4–12 months: 330 mg/d) and (**B**) phosphorus (0–3 months: 120 mg/day, 4–12 months: 180 mg/day) for term infants. Significant differences (adjusted *p*-value < 0.05) are marked with *.

**Figure 4 nutrients-16-03279-f004:**
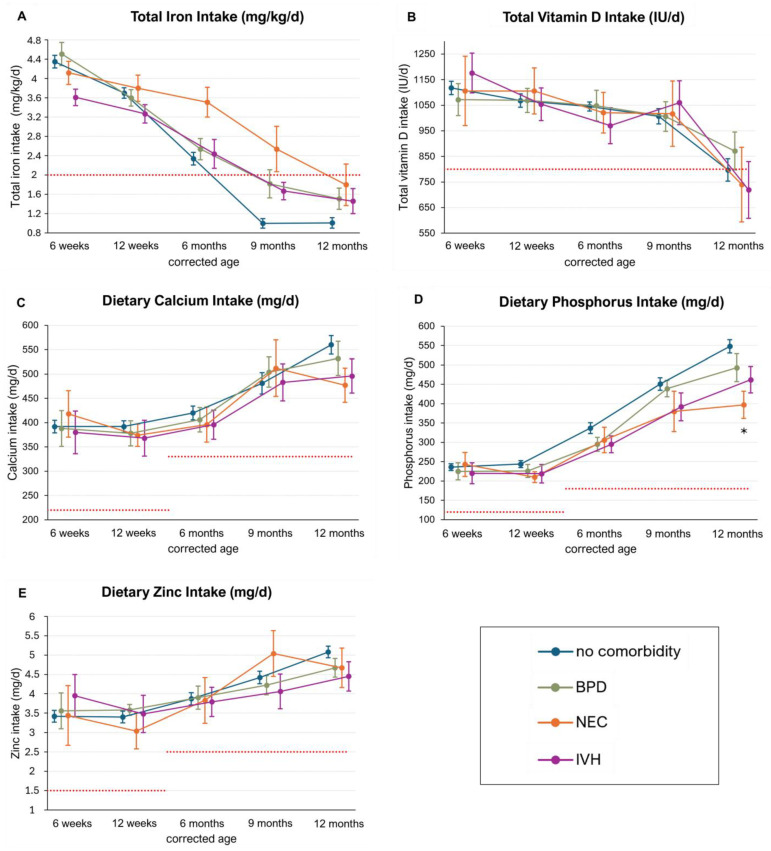
Micronutrient intakes in infants with and without comorbidities (BPD, NEC, IVH). Data are presented as the mean and standard error. Significant differences (adjusted *p*-value < 0.05) are marked with *. The red dotted lines represent the recommended dietary and supplemental intakes of the respective micronutrients. Total = dietary intake + supplementation. (**A**) Total Iron Intake (mg/kg/d) in infants with and without comorbidities. (**B**) Total vitamin D Intake (IU/d) in infants with and without comorbidities. (**C**) Dietary Calcium Intake (mg/d) in infants with and without comorbidities. (**D**) Dietary Phosphorus Intake (mg/d) in infants with and without comorbidities. (**E**) Dietary Zinc Intake (mg/d) in infants with and without comorbidities.

**Figure 5 nutrients-16-03279-f005:**
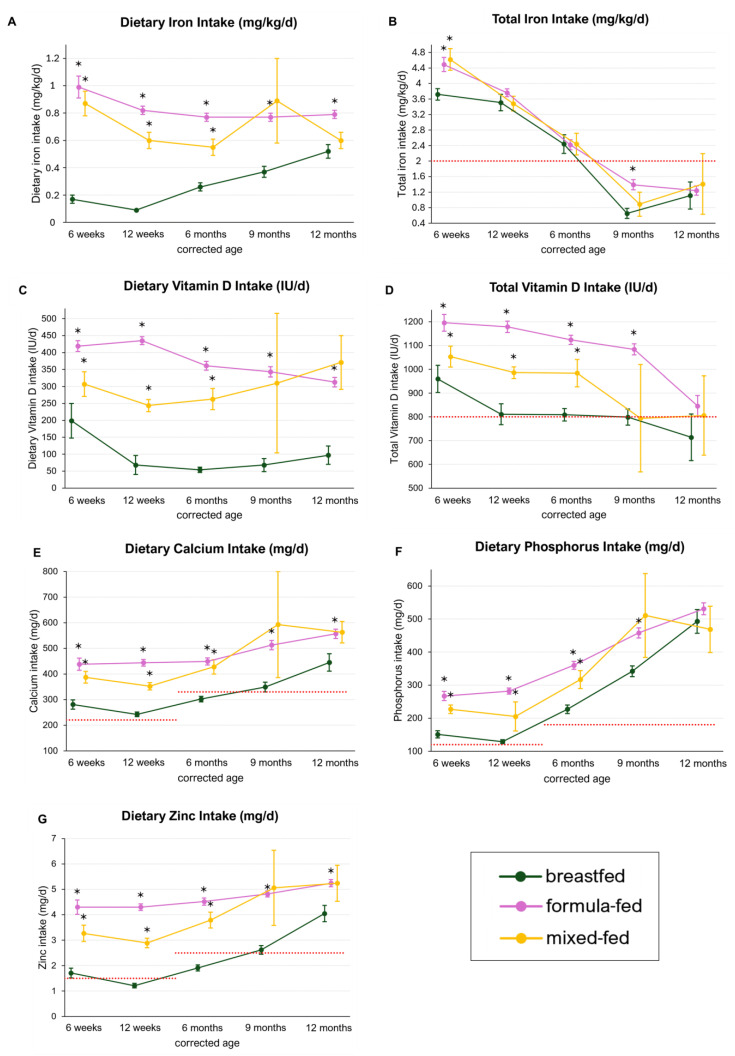
Micronutrients in breastfed, formula-fed, and mixed-fed infants. Data are presented as the mean and standard error. Statistical testing was performed comparing breastfed vs. formula-fed and breastfed vs. mixed-fed. Significant differences (adjusted *p*-value < 0.05) are marked with *. The red dotted lines represent the recommended dietary and supplemental intakes of the respective micronutrients. Total = dietary intake + supplementation. (**A**) Dietary iron intake (mg/kg/d) in breastfed, formula-fed and mixed-fed infants. (**B**) Total iron intake (mg/kg/d) in breastfed, formula-fed and mixed-fed infants. (**C**) Dietary vitamin D intake (IU/d) in breastfed, formula-fed and mixed-fed infants. (**D**) Total vitamin D intake (IU/d) in breastfed, formula-fed and mixed-fed infants. (**E**) Dietary calcium intake (mg/d) in breastfed, formula-fed and mixed-fed infants. (**F**) Dietary phosphorus intake (mg/d) in breastfed, formula-fed and mixed-fed infants. (**G**) Dietary zinc intake (mg/d) in breastfed, formula-fed and mixed-fed infants.

**Table 1 nutrients-16-03279-t001:** Dietary protocols valid for analysis.

CA	Early Group	Late Group	Total
6 weeks	86 (59%)	60 (41%)	146
12 weeks	106 (62%)	64 (38%)	170
6 months	90 (58%)	66 (42%)	156
9 months	63 (55%)	51 (45%)	114
12 months	65 (53%)	57 (47%)	122

CA: corrected age.

**Table 2 nutrients-16-03279-t002:** Baseline characteristics and nutrition.

Parameter	Early (n = 115)	Late (n = 82)	*p*-Value
Neonatal parameters			
Male sex	69 (60%)	36 (44%)	**0.04**
Gestational age at birth (days)	188 (±14)	187 (±17)	0.33
Gestational age (weeks + days)	26 + 6 (24 + 6–28 + 6)	26 + 5 (24 + 2–29 + 1)	0.33
Birth weight (g)	926 (±254)	881 (±262)	0.13
Neonatal morbidities			
Necrotizing enterocolitis ≥ grade II	5 (4%)	6 (7%)	0.42
Bronchopulmonary dysplasia	14 (12%)	23 (28%)	**0.01**
Intraventricular hemorrhage ≥ grade II	17 (15%)	12 (15%)	0.74
Periventricular leukomalacia	0 (0%)	1 (1%)	/
Nutrition			
Introduction of solid foods (weeks CA)	13.2 (± 3.0)	20.4 (± 2.9)	**<0.001**
Type of first solid food			
Vegetables	82 (71%)	54 (66%)	0.69
Fruits	17 (15%)	14 (17%)	0.82
Milk and dairy products	0 (0%)	1 (1%)	/
Cereals (with gluten)	1 (1%)	1 (1%)	/
Cereals (without gluten)	1 (1%)	2 (2%)	/
Several	11 (10%)	7 (9%)	0.99
Type of milk feeding at 6 weeks CA			
Breastfed	28 (24%)	34 (42%)	**0.01**
Formula-fed	66 (57%)	31 (38%)	**0.01**
Mixed-fed	18 (16%)	15 (18%)	1.00

Categorical data are presented as numbers with percentages in parentheses. Continuous data are presented as the mean and standard deviation in parentheses. CA: corrected age; *p*-values < 0.05 were considered statistically significant and marked bold.

## Data Availability

The study protocol and the individual participant data that underlie the results reported in this article, after de-identification, are available upon request from the corresponding author six months after publication. Researchers will need to state the aims of any analyses and provide a methodologically sound proposal. Proposals should be directed to nadja.haiden@kepleruniklinium.at. Data requestors will need to sign a data access agreement and, in keeping with patient consent for secondary use, obtain ethical approval for any new analyses due to ethical reasons.

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
