# Peer review of "Micronutrient Intake during Complementary Feeding in Very Low Birth Weight Infants Comparing Early and Late Introduction of Solid Foods: A Secondary Outcome Analysis"

_nutrients, 2024, doi:10.3390/nu16193279_

Round 1

Reviewer 1 Report (New Reviewer)

Comments and Suggestions for Authors

The impact of the timing of solid food introduction on micronutrient intake in very low birth weight (VLBW) infants remains unclear. In this observational study, micronutrient intake during complementary feeding in VLBW infants was investigated, categorized based on whether solids were introduced </≥17 weeks corrected age. Total vitamin D, calcium, zinc, and phosphorus greatly met intake recommendations, however, dietary iron intake was insufficient to equalize iron quantity from supplements during the second half year. While nutrient intakes were similar between infants with and without comorbidities, breastfed infants had lower micronutrient intakes compared to formula-fed infants. Consequently, the authors suggested that prolonged iron supplementation may be necessary beyond the introduction of iron-rich solids. In general, the study was well-performed and the manuscript was well-written.

Detailed comments:

1) Not all abbreviations have been introduced with full names at the first time of appearance, such as "p-adj." in Abstract, and "IU" in Line 113. What is "SE" after the Line 190?

2) Besides Table 1, how many infants were enrolled in this study should be indicated in the text of the Methods section.

3) How you got the grades of NEC or IVH should be introduced briefly in the Methods section.

4) I cannot understand "26+6 (±2+0)" and "26+5 (±2+2)" in Table 2. Whether they should be 32±2 and 31±4?

5) What does the red dotted line mean in Figure 5?

Comments on the Quality of English Language

Line 23: 24h --> 24-h

Line 25: expect --> except

There should be a space between the unit and the number, which should be checked throughout the text.

There are a large number of "Error! Reference source not found" throughout the text.

Author Response

We thank the reviewer for his/her close reading of our article, and for the constructive and detailed comments. 

Please see the point-by-point response in the attachment.

Kind regards.

Reviewer 2 Report (New Reviewer)

Comments and Suggestions for Authors

I would like to express my sincere gratitude for the opportunity to review your manuscript titled "Micronutrient Intake during Complementary Feeding in Very Low Birth Weight Infants comparing Early and Late Introduction of Solid Foods: A Secondary Outcome Analysis." It has been a pleasure to evaluate your work and provide constructive feedback to enhance the quality of your research.

The prospective observational design of your study, along with the comprehensive data collection methods and statistical analyses, demonstrates your commitment to conducting rigorous research. Your findings, which suggest that the timing of solid food introduction does not significantly affect micronutrient intake, with the exception of iron and phosphorus at 6 months corrected age, contribute to the growing body of knowledge in this field. The subgroup analyses comparing infants with and without comorbidities and the comparison of breastfed, formula-fed, and mixed-fed infants add depth to your study and provide important insights for clinical practice. I have only some suggestions:

Title: The title accurately reflects the content of the study, clearly indicating the focus on micronutrient intake in very low birth weight (VLBW) infants during complementary feeding. However, the title could be more concise by removing the redundant phrase "Part II."

Abstract:

- The abstract provides a clear and concise summary of the study's background, methods, results, and conclusions.

- The objectives and main outcomes are well-defined, allowing readers to quickly grasp the study's focus and findings.

- Consider including the sample size in the methods section of the abstract to provide context for the results.

Introduction:

- The introduction effectively highlights the importance of the complementary feeding phase for addressing micronutrient imbalances in VLBW infants.

- The authors provide a solid rationale for the study by identifying the research gaps in the current literature, particularly regarding micronutrient intake in a representative VLBW population.

- The objectives of the study are clearly stated and align with the identified research gaps.

Methods:

- The study design, a prospective observational study, is appropriate for the research objectives.

- The inclusion and exclusion criteria are well-defined, ensuring a representative VLBW cohort.

- The data collection methods, including the use of 24-hour recalls and 3-day dietary records, are suitable for assessing micronutrient intake.

- The statistical analysis plan is comprehensive and appropriate for the study design and objectives.

- Consider providing more details on the sample size calculation and justification for the chosen sample size.

Results:

- The results section is well-organized, presenting the findings in a logical sequence.

- The authors provide clear and concise descriptions of the main findings, supported by appropriate statistical analyses.

- The tables and figures effectively complement the text, providing visual representations of the key results.

- The subgroup analyses of infants with and without comorbidities and the comparison of breastfed, formula-fed, and mixed-fed infants add depth to the study and provide valuable insights.

Discussion:

- The discussion section thoroughly interprets the main findings, placing them in the context of the current literature.

- The authors provide plausible explanations for the observed results and discuss the potential implications for clinical practice.

- The strengths and limitations of the study are adequately addressed, demonstrating a balanced and critical evaluation of the research.

- The conclusions are supported by the study's findings and provide clear recommendations for future research and clinical practice.

Overall, this study makes a valuable contribution to the understanding of micronutrient intake during complementary feeding in VLBW infants. The authors have conducted a well-designed and executed study, and the manuscript is well-written and organized.

Suggestions for improvement:

1. Consider streamlining the title by removing "Part II" to improve clarity and conciseness.

2. Include the sample size in the methods section of the abstract to provide context for the results.

3. Provide more details on the sample size calculation and justification in the methods section.

4. Discuss the potential impact of the study's findings on current clinical guidelines and recommendations for micronutrient supplementation in VLBW infants.

5. Expand on the implications of the study's findings for infants with comorbidities, highlighting the need for future research to establish specific micronutrient requirements for these subgroups.

In conclusion, this study is a well-conducted and valuable contribution to the field of neonatal nutrition. The authors have effectively addressed an important research gap and provided insights that can inform clinical practice and future research. 

Comments on the Quality of English Language

Minor editing

Author Response

We thank the reviewer for his/her close reading of our article, and for the constructive comments. 

Please see the point-by-point response in the attachment. 

Kind regards. 

Reviewer 3 Report (New Reviewer)

Comments and Suggestions for Authors

The authors present the results of a secondary outcome analysis of micronutrient intake during complementary feeding in VLBW infants. This study was well conducted and the data appropriately analysed. I should like to suggest two minor amendments.

1. Line 161. Please state the version of the IDE R Studio.

2. Lines 194, 212, 295, 301 and 304.

Author Response

We thank the reviewer for his/her close reading of our article, and for the constructive comments. 

Please see the detailed point-by-point response in the attachment. 

Kind regards. 

This manuscript is a resubmission of an earlier submission. The following is a list of the peer review reports and author responses from that submission.

Round 1

Reviewer 1 Report

Comments and Suggestions for Authors

Nutrients-2065343

This article reports the results of Nutrient Intake in Preterm Infants on Early Solid Foods during the First Year of Life: A Secondary Outcome Analysis of a Prospective, Randomized Intervention Study. The presented results are good and well-managed citing the proper literature. However, some points need further improvements, and additional data should be supplied to improve the quality of the manuscript. Therefore, I recommend that this paper should be considered for publication after following major revision.

1.       In the introduction part, the logicality is not very explicit, and there are some English grammatical errors; please double-check and correct them.

2.       Authors are encouraged to cite some relevant and recent literature and improve the concept of the paper and highlight the novelty of the current work, such as.: International Journal of Molecular Sciences 20.9 (2019): 2206. and   IOP Conference Series: Earth and Environmental Science. Vol. 346. No. 1. IOP Publishing, 2019., In order to strengthen their ideas in the light of previous work.

3.       It would be appreciated if the authors could draw a table for a better comparison in the main text.

4.       The optimization and efficiency of the different kinds of species should be discussed in more detail.

5.       All equations should be checked and modified in math-type software.  

6.       The resolution of Fig. 1 and 3 should be enhanced.

Reviewer 2 Report

Comments and Suggestions for Authors

Nutrition of preterms is a largely debated subject. There are some general recommendations for adequate alimentation for this group of patients.

Alimentation during the first 1000 days has a very strong impact on well-being later in life. Inadequate protein intake can cause obesity and related comorbidities. High glucose intake will increase the risk of diabetes.

Your study is interesting but needs more patients to be enrolled just to be statistical significant.

The conclusion section should be more elaborate and should emphasize the role of this paper in preterm babies.

Round 2

Reviewer 1 Report

Comments and Suggestions for Authors

Revision is not satisfactory; many questions are remaining to provide suitable answers.

I recommend authors should do revision seriously. At this stage article has serious flaws, additional experiments are needed, and research is not conducted correctly.